# Advanced Techniques for Pilotis RC Frames Seismic Retrofit: Performance Comparison for a Strategic Building Case Study

**Eleonora Grossi \*** , **Matteo Zerbin and Alessandra Aprile**

Engineering Department, University of Ferrara, via Saragat 1, 44122 Ferrara, Italy; matteo.zerbin@unife.it (M.Z.); alessandra.aprile@unife.it (A.A.)

\* Correspondence: eleonora.grossi@unife.it

**Abstract:** Pilotis buildings have widely spread out in developed countries since World War II onwards. From the structural point of view, Pilotis RC frames exhibit substantial lack in ductility capacity and shear resistance localized at the first floor, since they have been mainly realized before the seismic codes' era. The present study shows the performance comparison of four advanced retrofit techniques when applied to typical Pilotis RC frame designed for gravity loads only according to Italian building code of '60s. A preliminary investigation has been performed to select non-linear numerical models suitable to describe the considered RC frame behavior, involving flexural inelastic hinges of RC beams and columns and in-plane axial inelastic hinges of masonry infill panels. Two seismic retrofit projects have been designed at a local level, by strengthening the masonry infilled panels with Fiber Reinforced Cementitious Matrix (FRCM) technique and alternatively by replacing infilled panels with prefabricated panels disconnected from the structure, so that no infill/frame interaction occurs. Two more retrofit projects have been designed at a global level, in order to improve the overall structural performance making use of energy dissipation and, alternatively, base isolation techniques. Nonlinear time history analysis and structural assessment have been carried out for the as-built case as well as for the four retrofit solutions according to Eurocode 8 and Italian Building Code, in order to highlight the structural deficiencies and relative improvements, respectively. Performances offered by the proposed retrofit techniques have been finally compared in terms of structural behavior, expected damage, and economic impact.

**Keywords:** advanced seismic retrofit techniques; Pilotis RC frame; seismic performance and cost-benefit analysis

## 1. Introduction

Pilotis-type building was firstly introduced in 1923 by Le Corbusier, one of the most important modern architects, as an RC residential building whose first-floor was designed to be a wide open space, isolating the structure from the ground level [1]. This modern conception, together with the development of reinforced concrete technology during the '50–'60s, led to the erection of a great number of Pilotis RC frames in Italy and in other parts of the world. However, seismic awareness was only at the beginning and the design of most of these buildings was carried out for gravity loads only, bringing to an increase of vulnerable built heritage [2].

Typical structural deficiencies of Pilotis buildings consist of poor ductility capacity and shear resistance of columns, as clearly put in evidence during the San Fernando Earthquake (1971), where the soft-storey effect led to the collapse of the Olive View Hospital main building [3] and the Mexico City's Earthquake experience (1985) [4,5]. Nowadays, it is well known how masonry infilled panels of

RC frames can experience in-plane failure or out-of-plane expulsion under the effect of earthquake loads, as also occurred in recent Central Italy Earthquakes of L'Aquila (2009) and Amatrice (2016) [6,7]. The effects of masonry infill panels collapse are reflected not only on the building itself [8], but on the nearby areas, increasing the damage incidence [9,10] and repair costs [11].

While modern design codes take in account strength and ductility capacity of both structural and non-structural elements, in earthquake prone geographical areas there is a great number of existing Pilotis buildings that still need urgent retrofit interventions due to severe structural deficiencies. Several studies have been published concerning the seismic retrofit of existing RC frames, comparing different rehabilitation strategies and including cost-benefit analysis; among the others, see [12–15]. However, thorough studies that are focused on Pilotis RC frames are not known to the authors.

An accurate assessment of the as-built and retrofit structural configurations can be obtained by using non-linear numerical models, suitable to describe the inelastic behavior of frame elements, and of in-plane masonry panels. The importance of non-linear time history analysis and advanced simplified design method, such as the Direct Displacement Based Design (DDBD) [16] or updated traditional Force-Based Design (FBD), to define the correct structural performance under earthquake actions was documented by many authors; among the others, see [17–19]. It has been emphasized how traditional FBD usually overestimates seismic actions, particularly for existing structures, due to the need of a preliminary selection of the behavior factor q, which plays a significant role [20].

The main purpose of this study is to develop a design strategy for the seismic retrofit of Pilotis RC frames, in order to select the most suitable advanced technique from both the structural performance and the intervention cost point of view. Additionally, among the many inelastic numerical models available in the literature, the most efficient modelling approach is selected in order to support professionals in facing time-requiring computational efforts and conceptual complexity related to the application of non-linear analysis and advanced design methods.

This paper presents the case study of a strategic Pilotis RC frame hosting essential functions and located in Italy. After a preliminary discussion about the proper implementation of non-linear models using a commercial software, a non-linear time history analysis of the as-built configuration is carried out in order to assess structural vulnerabilities, according to Italian Building Code (NTC 2018) [21] and Eurocode 8 [22] requirements. Subsequently, four different retrofit projects are proposed, based on advanced retrofit techniques. Two retrofit projects operate interventions at the local level by strengthening masonry infilled panels with the Fiber Reinforced Cementitious Matrix (FRCM) technique and alternatively by replacing infilled panels with light prefabricated panels disconnected from the structure in order to prevent the panel/frame interaction. FRCM technique has been widely investigated during the last decades for the strengthening of masonry walls in both in and out-of-plane directions [23–28]. On the other hand, infill panels disconnection from the structure has proved to be an efficient way to remove the typical Pilotis behavior [29–31]. Two more retrofit projects operate interventions at the global level by improving the overall structural performance by means of Friction Dampers (FD) [32,33] and, alternatively, Lead Rubber Bearings (LRB) [34] applications. Energy Dissipation and Base Isolation techniques have been developed in Italy for many years [35] but they have found extensive application in rehabilitations projects [36–39] after the inclusion of such innovative retrofit techniques in the 2008 Building Code [40] upgrade. Nonlinear time history analysis is also carried out for the retrofitted structural configurations to evaluate the obtained structural performances. Additionally, intervention costs analysis is performed and presented. Finally, improvements that are offered by the proposed retrofit techniques are compared in terms of structural behavior, expected damage, and economic impact.

## 2. Case Study

This study has been conducted for a typical pilotis RC frame system designed for gravity loads only during the decade '60–'70s in Italy, with a 12.45 × 36 m rectangular base, and seven floors for a total of 22 m high. Structural plan and views are presented in Figure 1 while RC materials information are reported in Table 1, as derived by a preliminary on-site investigation. Main beams

span in transversal direction (named X direction) while only perimetric secondary beams connect main beams in longitudinal direction (named Y direction). The building was first conceived to host essential functions, including public administration functions and directional offices: this led to different interstorey heights, as highlighted from the frontal views of Figure 1b,c.

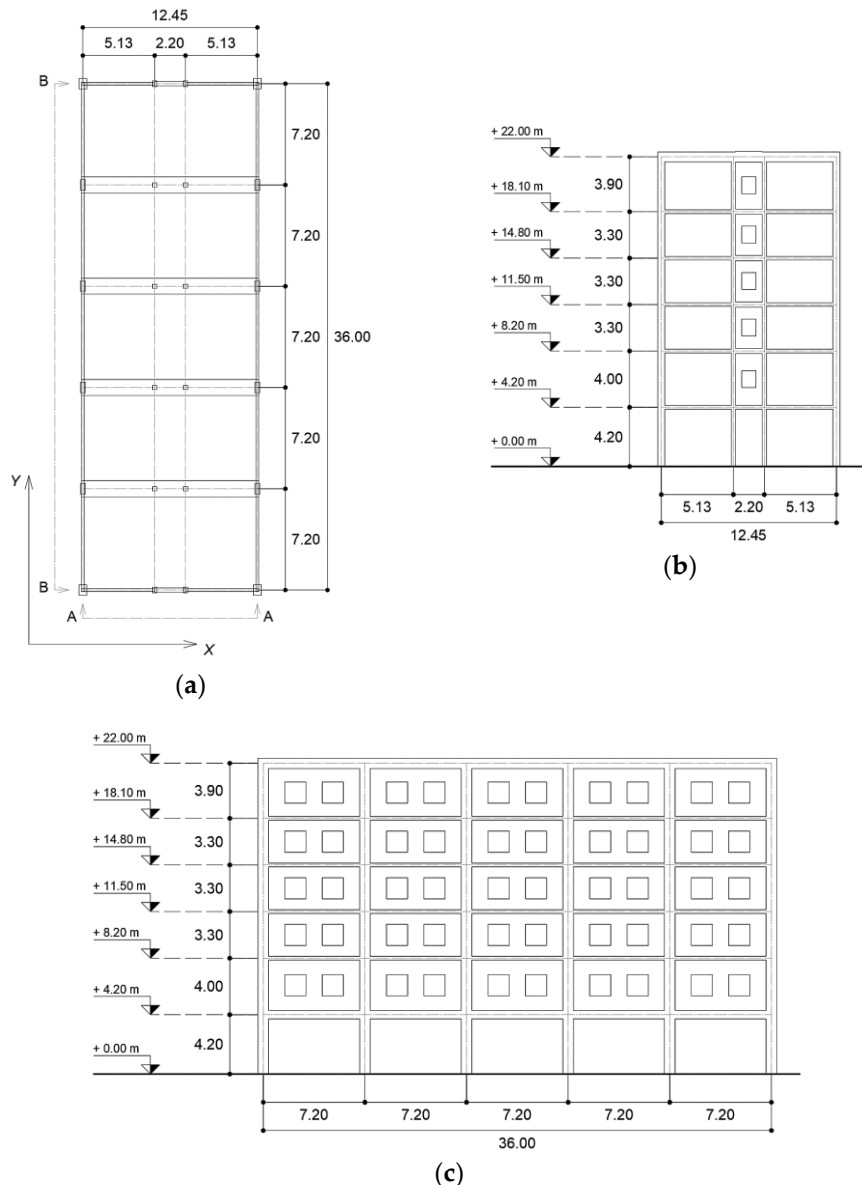

**Figure 1.** Pilotis RC frame (**a**) plan, (**b**) A-A view, and (**c**) B-B view. Dimensions in m.

**Table 1.** Material properties for RC frame.

| f$_{cm}$ [MPa] | E$_c$ [MPa] | f$_{ym}$ [MPa] |
|:---:|:---:|:---:|
| 22.4 | 28018 | 409 |

A nonlinear time history analysis has been performed in order to accurately evaluate the case study structural performance under seismic action. The considered building is located in one of Italian highest seismic risk zones, characterized by a Peak Ground Acceleration (PGA) of 0.36 g and Soil Amplification Factor of 1.17. A group of seven natural ground motions with two components have been selected, whose spectral compatibility have been checked by comparison of the mean spectrum with the target elastic spectrum, as reported in Figure 2a,b for, respectively, X and Y directions.

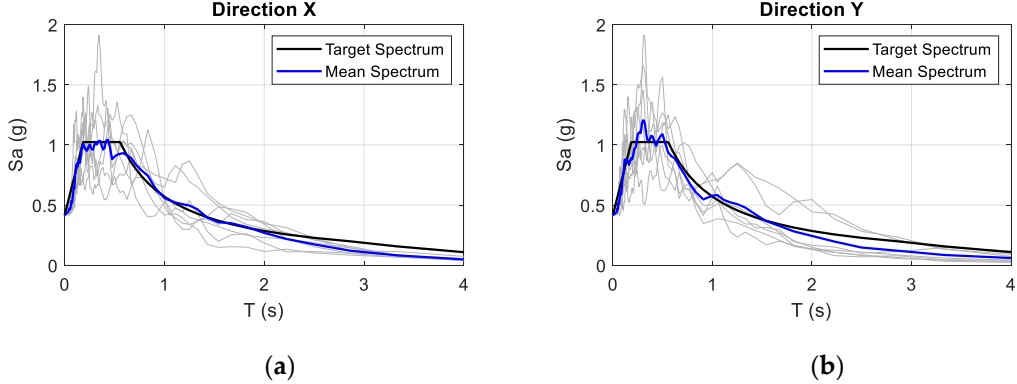

**Figure 2.** Natural ground motions mean spectrum and target spectrum of the selected site for (**a**) X and (**b**) Y directions.

## 2.1. Non-Linear Models

A preliminary investigation has been performed to select non-linear numerical models that are suitable to describe the considered RC frame behavior, involving flexural inelastic hinges of RC beams and columns and in-plane axial inelastic hinges of masonry infill panels.

The non-linear numerical analyses have been carried out using the commercial FEM code Midas/Gen [41], while MATLAB [42] software was used to pre- and post-process data.

### 2.1.1. Beams and Columns Flexural Inelastic Hinges

Idealized inelastic models of beam and column elements stand out by the way their plasticity evolution is distributed along the element length, defining two main groups: concentrated (or lumped) and distributed plasticity models.

The first group models concentrate the inelastic behavior at the end of the elements using a moment-rotation law through a rigid-plastic hinge or a inelastic spring with hysteretic properties, as outlined, respectively, by Figure 3a,b. The second group models are progressively more complex than the previous ones. In the finite length hinge model pointed out in Figure 3c, the plasticity formulation is designated while using a moment-curvature law at the end of the element through a hinge zone of finite length. In the fiber section model figured in Figure 3d, the plasticity is distributed by numerical integration all along member length, defining non-linear axial stress-strain hysteretic laws for the cross section materials. Finally, in the last and most complex model shown in Figure 3e, a continuum discretization is implemented along the whole member through finite elements with non-linear constitutive properties, leading to a tricky parameter calibration and demanding computational efforts [43].

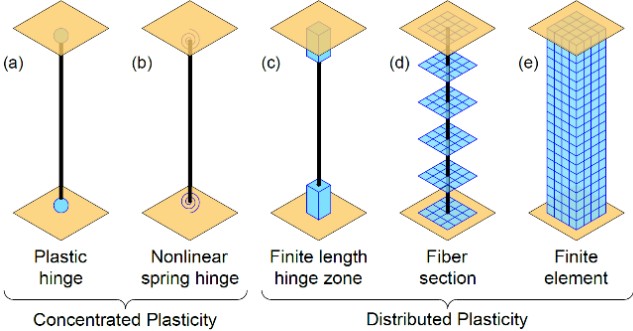

**Figure 3.** Idealized inelastic models of beams and columns, implemented as (**a**) plastic hinge, (**b**) nonlinear spring hinge, (**c**) finite length hinge zone, (**d**) fiber section and (**e**) finite element.

Non-linear spring (NS), finite length hinge zone (FL) and fiber section (FS) models have been considered in the preliminary investigation in order to select a simple but efficient numerical

implementation technique for RC frame nonlinear modelling. NS and FL models are easily carried out starting from the geometric definition of the cross sections and assuming nonlinear constitutive laws of materials, as defined by Eurocode 8 [22] and NTC 2018 [21]. As for FS models is concern, axial stress-strain laws have been defined using Kent-Park and Menegotto-Pinto formulations [44,45], for concrete and steel bars, respectively, and four integration points for a better convergence [46]. NS, FL and FS models have been implemented for the considered RC frame and worked out using a non-linear time history analysis. In Figure 4, the NS, FL, and FS hysteretic cycles obtained for a column and a beam have been compared to the backbone (B) curves representing the analytical moment-curvature law of the selected sections. The B curve has been carried out for the geometry and rebars' configuration of the selected cross sections using a Midas/Gen [41] software tool, according to Eurocode 8 [22] and NTC 2018 [21].

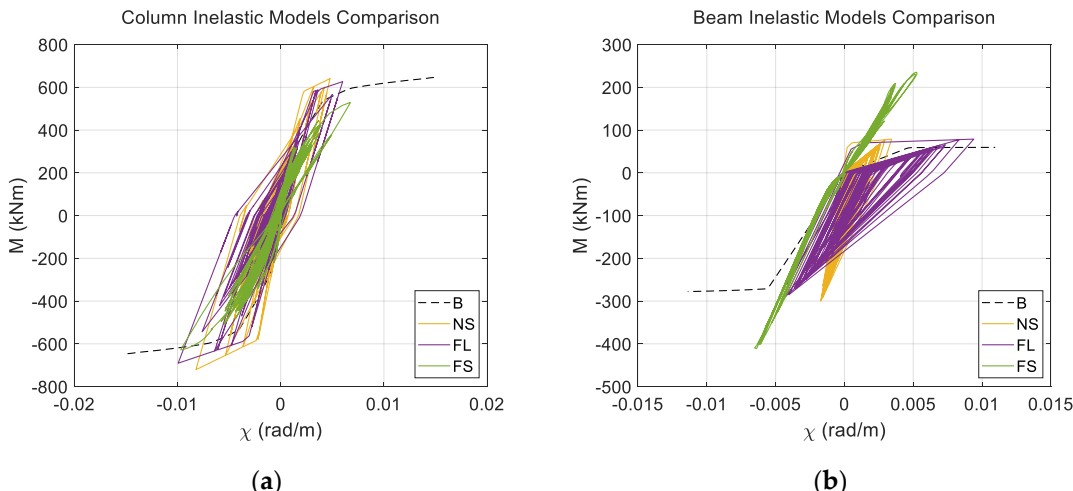

**Figure 4.** Inelastic models comparison for (**a**) columns and (**b**) beams elements.

Concerning the column modelling, NS, FL, and FS models' curves are in good agreement with the backbone one. The FS model provides the most accurate curve, while FL and NS models provide curves with higher stiffness when compared to the backbone one, even if a good strength prediction is observed. Concerning the beam modelling, FS model strength and NS model stiffness are much higher than the backbone one.

This last remark, as well as the numerical convergence issues found out during the FS model implementation, led to the selection of FL model to be implemented in this study for beams and columns. In fact, the FL model provides a good compromise in terms of accuracy and stability of the numerical solution.

2.1.2. Masonry Infill Panels In-Plane Models Axial Hinges

Masonry infill panels are usually modelled by diagonal compressed equivalent struts, implemented as linear truss elements using Stafford–Smith theory [47,48] and Al Chaar theory to account for openings and previous damage [49], as largely transposed by the NTC 2018 [21]. The equivalent strut has same thickness $t$ of the selected panel, while its width $a$ is computed as a function of the adimensional parameter $\lambda_1 H$ defined, as follows:

$$\lambda_1 H = H\left(\frac{E_m t sin2\theta}{4E_c Jh}\right)^{1/4} \tag{1}$$

where:

- $H$, $h$, $t$ and $\theta$ are geometric parameters showed in Figure 5,

- $J$ is the moment of inertia of confining column,
- $E_m$ is masonry elastic modulus, and
- $E_c$ is concrete elastic modulus.

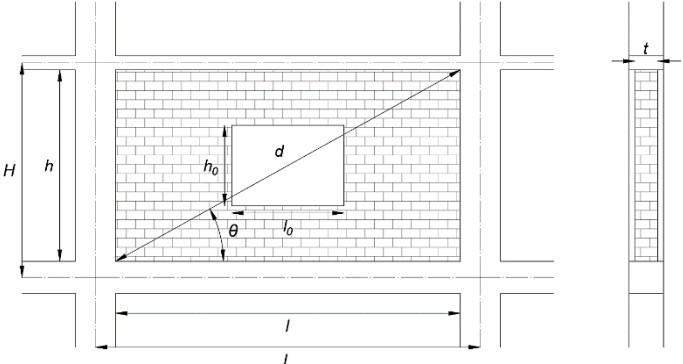

**Figure 5.** Infill panel geometric properties.

The equivalent width $a$ is given by Equation (2) for whole panels, while a reduced equivalent width $a_{red}$ is given by Equation (3) when some section reduction can affect the infilled panel behavior:

$$a = 0.175d(\lambda_1 H)^{-0.4} \tag{2}$$

$$a_{red} = a\ R_1\ R_2 \tag{3}$$

where $d$ is panel diagonal length and $R_1$, $R_2$ are the reduction factors that take into account openings and previous damage as defined by Al Chaar [49].

The maximum horizontal force $H_{max}$ carried by infill panels can be evaluated as:

$$H_{max} = min(H_{0,i})\ with\ i\ =\ 1, 3 \tag{4}$$

$H_{0,i}$ are the maximum horizontal forces related to typical failure modes, reported in Table 2, where $f_{vk0}$ and $f_k$ are, respectively, initial shear strength under zero compressive stress and compressive strength of masonry panels.

**Table 2.** Stafford-Smith Failure Modes adapted from [47,48].

| | |
|---|---|
| Bed Joints Sliding | $H_{0,1} = f_{vk0}\sqrt{1 + \dfrac{(0.8\frac{h}{l}-0.2)H_{0,BS}}{1.5f_{vk0}lt}}\,lt$ |
| Diagonal Tensile Failure | $H_{0,2} = \dfrac{f_{vk0}}{0.6}lt$ |
| Corner Crushing | $H_{0,3} = 0.8f_k cos^2\theta\sqrt[4]{\dfrac{E_c}{E_m}Jht^3}$ |

This modelling approach is very effective for linear analysis only, due to the lack of cracking and post-failure behavior.

Starting from the Stafford–Smith linear formulation, Decanini et al. model [50,51] developed an inelastic equivalent strut model, whose equivalent strut width $\omega$ is given by Equation (5):

$$\omega = \left(\frac{K_1}{\lambda H} + K_2\right)d \tag{5}$$

where $\lambda H$ is given by Equation (1) and coefficients $K_1$, $K_2$ are reported in Table 3 as a function of $\lambda H$.

**Table 3.** Coefficients $K_1$ and $K_2$.

|  | $K_1$ | $K_2$ |
|---|---|---|
| $\lambda H < 3.14$ | 1.3 | −0.178 |
| $3.14 \leq \lambda H \leq 7.85$ | 0.707 | 0.01 |
| $7.85 < \lambda H$ | 0.47 | 0.04 |

The maximum horizontal force $H_{max}$ carried by infill panels can be evaluated as:

$$H_{max} = min\left(\sigma_{br,i}\right) t\omega \, cos\theta \text{ with } i = 1, 4 \tag{6}$$

$\sigma_{br,i}$ are the maximum axial stresses related to typical failure modes reported in Table 4, where:

- $\sigma_0$ is the vertical stress, set to 0 MPa due to the lack of vertical load acting on the panels,
- $\sigma_{m0}$ is the compression strength,
- $\tau_{m0} = \gamma \sqrt{\sigma_{m0}}$ is the shear strength, with $\gamma = 0.6 \div 1.5$ here assumed equal 1, and $\sigma_{m0}$ expressed in (kg/cm$^2$), and
- $u = 0.7\tau_{m0}$ is the sliding strength.

**Table 4.** Decanini et al. Failure Modes adapted from [50,51].

| Bed Joints Sliding | $\sigma_{br,1} = \dfrac{(1.2\,sin\theta + 0.45\,cos\theta)u + 0.3\sigma_0}{\omega/d}$ |
|---|---|
| Diagonal Tensile Failure | $\sigma_{br,2} = \dfrac{0.6\tau_{m0} + 0.3\sigma_0}{\omega/d}$ |
| Corner Crushing | $\sigma_{br,3} = \dfrac{1.12\,sin\theta\,cos\theta}{K_1(\lambda H)^{-0.12} + K_2(\lambda H)^{0.88}}$ |
| Diagonal Compressive Failure | $\sigma_{br,4} = \dfrac{1.16\sigma_{m0}\,tan\theta}{K_1 + K_2\lambda H}$ |

The secant stiffness contribution $K_{sec}$ of the equivalent strut in horizontal direction is defined by Equation (7). In order to consider the strength reduction due to openings, both $H_{max}$ and $K_{sec}$ must be multiplied for a coefficient $\rho_0$ reported in Equation (8), where $\alpha_a = (l_0 h_0)/(lh)\cdot 100$, $\alpha_1 = (l_0)/(l)\cdot 100$ and $l_0, h_0, l, h$ are defined in Figure 5.

$$K_{sec} = \frac{E_m t\omega}{d}cos^2\theta \tag{7}$$

$$\rho_0 = 0.55e^{-0.035\alpha_a} + 0.44e^{-0.025\alpha_1} \tag{8}$$

Figure 6 shows a graphical representation of the non-linear constitutive model by Decanini et al. [50,51], where the expected drifts at cracked $\delta_{cr}$, maximum $\delta_{max}$, and residual $\delta_{res}$ stages are empirically obtained starting from initial $K_i = 4K_{sec}$ and residual $K_{res} = 0.02\,K_{sec}$ stiffness values and horizontal forces at crack $H_{cr} = 0.8\,H_{max}$ and residual $H_{res} = 0.35\,H_{max}$ stages.

Further studies showed that the Decanini et al. [50,51] constitutive model can correctly predict the infill panel experimental behavior in term of strength, but not in terms of equivalent stiffness, which results in being overestimated [52]. Cardone, Perrone, and Sassun [53] recently proposed a Decanini modified model, revising the expected panel drift $\delta$, as follows:

$$\delta = \frac{L}{h} - \sqrt{(1-\varepsilon)^2\left[1 + \left(\frac{L}{h}\right)^2\right] - 1} \tag{9}$$

where $\varepsilon$ is the diagonal strain capacity of the considered infill panel.

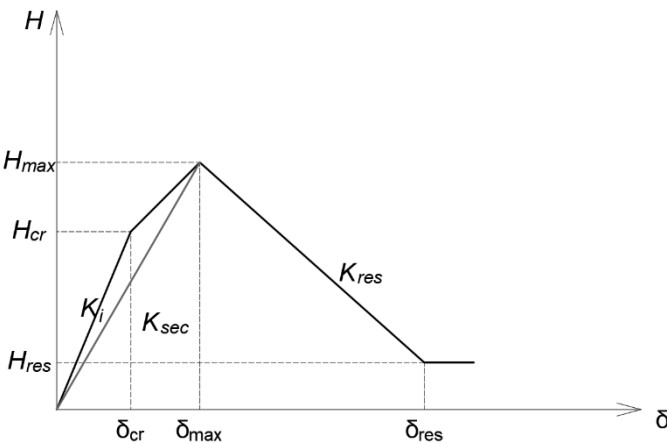

**Figure 6.** Infill panels backbone curve adapted from Decanini et al. [50,51].

A comparison has been performed for a masonry panel whose geometry and material properties are listed in Tables 5 and 6, respectively, in order to select the model that better describes the infill panels behavior for the considered case study. Masonry elastic modulus and compressive strength are selected from previous on-site investigations, while shear strengths and strain capacities $\varepsilon$ are gathered from the average values of experimental results reported by Cardone et al. [53], which investigated both weak and strong masonry infill panels.

**Table 5.** Panel geometry.

| $H$ [m] | $h$ [m] | $L$ [m] | $l$ [m] | $t$ [m] | $h_0$ [m] | $l_0$ [m] | $\theta$ |
|---------|---------|---------|---------|---------|-----------|-----------|----------|
| 4 | 3.72 | 5.125 | 4.7 | 0.16 | 0 | 0 | 38.4° |

**Table 6.** Panel material properties for selected models.

| | | |
|---|---|---|
| $E_c$ [MPa] | 27,938 | |
| $E_m$ [MPa] | 5000 | |
| $f_k$ [MPa] | 5.50 | Strength for Stafford Smith [47,48] |
| $f_{vk0}$ [MPa] | 0.37 | |
| $\sigma_0$ [MPa] | 0 | |
| $\sigma_{mo}$ [MPa] | 5.50 | Strength for Decanini et al. [50,51] |
| $\tau_{mo}$ [MPa] | 0.74 | |
| $u$ [MPa] | 0.52 | |
| $\varepsilon_{cr}$ [%] | 0.09 | |
| $\varepsilon_{max}$ [%] | 0.2 | Strain for Decanini Modified [52,53] |
| $\varepsilon_{res}$ [%] | 0.9 | |

The obtained results are showed in Figure 7 where the horizontal force $H_m$ applied to the panel is plotted as a function of the top horizontal displacement $d$.

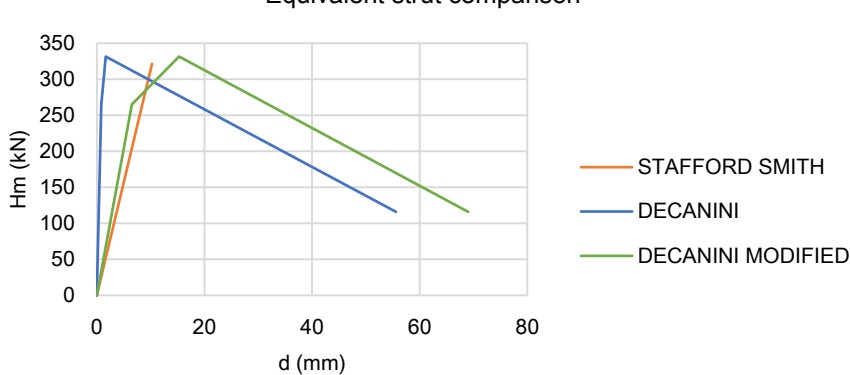

**Figure 7.** Comparison of infill panel equivalent strut constitutive models.

The difference between Decanini [50,51] and Decanini modified [52,53] models in terms of initial stiffness and top horizontal displacement is clearly visible, while in terms of maximum strength, all the selected models show similar results. It can be observed that Stafford–Smith model [47,48] elastic behavior is very similar to the one of Decanini modified model [52,53]. This last remark, in addition to the accredited prediction efficiency of the Stafford–Smith model, has led to the choice of the Decanini modified model [52,53] for the implementation of masonry infill panels of this study.

The infill panels constitutive models obtained for the case study are finally reported in Figure 8, while Figure 9 represents their location and relative identification colors. Infill panels have been dived into groups that consider interstory heights and panels geometry as well, obtaining the nine different acronyms of Figure 8. F1, F2, and F3 characterize three different interstorey heights: F1 is used for the second storey height, F2 for the third, fourth, and fifth storey heights, and F3 for the sixth storey height. A and B characterize the transversal and longitudinal building sides, as shown by Figure 1b,c, respectively. Lastly, L and C characterize lateral and central infill panels, respectively, of the transversal building side.

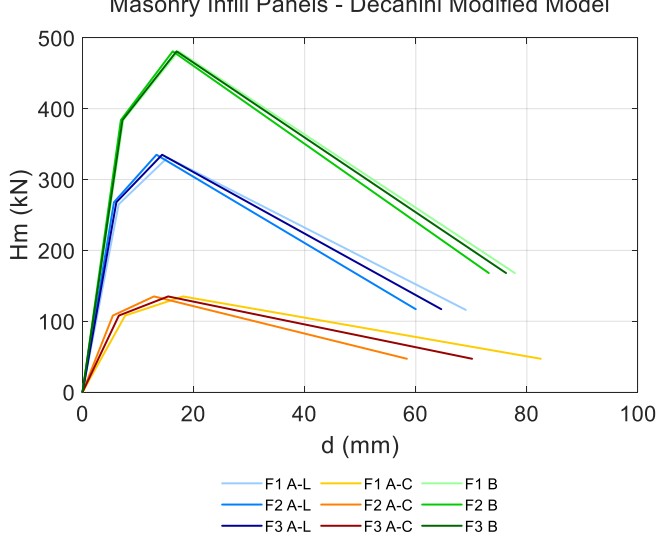

**Figure 8.** Case study masonry infill panels constitutive models.

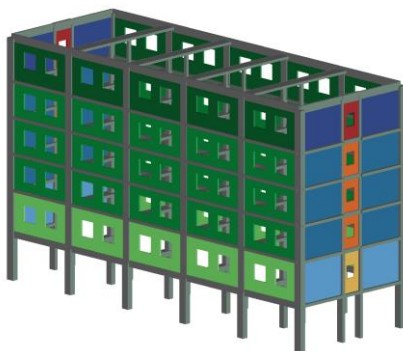

**Figure 9.** Case study representation with masonry infill panels IDs location.

### 2.2. Performance Assesment Criteria

Performance assessment criteria have been defined according to NTC 2108 [21], Circ. C.S.LL.PP. 2019 [54], and Eurocodes [22]. RC beams and columns have been checked for brittle failure modes, comparing shear demand $V_{Ed}$ with capacity $V_{Rd}$, and for ductile failure modes, comparing the element chord rotation demand $\theta_{Ed}$ with capacity $\theta_u$. The infill panels have been checked for out-of-plane and in-plane failure modes, by comparing the out-of-plane flexural moment demand $M_{Ed}$, with capacity $M_{Rd}$, and the in-plane horizontal displacement demand $d_{Ed}$ with capacity $d_u$, respectively. Performance ratios, $pr$, have been defined for each type of crisis to easily evaluate the structural behavior. Shear performance ratios $pr_V$ and rotational performance ratios $pr_\theta$ have been defined for RC elements, according to (10) and (11), while moment performance ratios $pr_M$ and displacement performance ratios $pr_d$ have been defined for infill walls, according to (12) and (13). The safety check is positive whenever $pr$ values are lower than 1.

$$pr_V = \frac{V_{Ed}}{V_{Rd}} \tag{10}$$

$$pr_\theta = \frac{\theta_{Ed}}{\theta_u} \tag{11}$$

$$pr_M = \frac{M_{Ed}}{M_{Rd}} \tag{12}$$

$$pr_d = \frac{d_{Ed}}{d_u} \tag{13}$$

Frame elements and infill panels demands have been set performing nonlinear time history analysis using the group of seven natural records of two components shown in Figure 2, selected for the Severe Damage (SD) Limit State of the considered site according to NTC 2018 [21]. Elements chord rotation capacity $\theta_u$ has been set according to Circ. C.S.LL.PP. 2019 [54] and Eurocodes [22] for the SD Limit State. The infill panels in-plane horizontal displacement capacity $d_u$ has been set according to the maximum displacement associated to panels strength $H_{max}$, as assessed by Decanini modified model [52,53]. Unconfined beam-to-column joints have been also checked in terms of tensile and compressive strength according to Par. C8.7.2.3.5 of Circ. C.S.LL.PP. 2019 [54].

The overall structural performance is finally evaluated according to the Italian DM 65/2017 [55], also called "Sismabonus", which classifies the seismic risk level of a structure from the worst class, G, to the best class, A+, using the Safety Index (SI) and Expected Annual Loss (EAL) concepts. SI, expressed as $\alpha$ in the following, is defined as ratio between the capacity peak ground acceleration, $PGA_C$, associated to the limit structural behavior, and the demand peak ground acceleration, $PGA_D$, associated to the considered Limit State. $PGA_C$ is obtained rescaling the group of selected accelerograms until the conducted safety checks give positive results, while the $PGA_D$ is the target spectrum PGA for the considered site. EAL is defined as the average annual amount of damage, associated to the repairing costs due to future expected earthquakes, taking in account the frequency and severity of all

the possible earthquakes compatible with the seismic hazard of the site [15], as given by NTC 2018 [21] standard for new constructions. The seismic risk level of the structure is assessed as the lower level between the SI and EAL-based risk levels.

It is to remark that the "Sismabonus" [55] is an innovative Italian Government law conceived to reduce the seismic risk level of the Italian built heritage by supporting the buildings' owners from the economical point of view with a tax break between 50% and 85% of the structural retrofit intervention costs over five years, depending on the seismic risk level that was obtained for the retrofit structure. With this law, the Italian Government actively means to spread the seismic risk awareness among citizens.

## 2.3. As-Built Performance Assesment

Non-linear time history analysis has been conducted of As-Built (AB) structural configuration while using the group of seven natural records of two components that are shown in Figure 2, selected for the SD Limit State of the considered site according to NTC 2018 [21]. As already reported in Section 2.1, the structure inelastic behavior has been numerically implemented using inelastic hinges of finite length for RC beams and columns, while using compression only inelastic links for masonry infill panels, according to Decanini Modified Model [28,29], as shown in Figure 8.

AB storey shear forces and displacements are plotted in Figure 10 charts for both X and Y direction. Maximum storey shear forces are 3500 kN and 5500 kN in X and Y directions, respectively. The charts show that infill panels absorb from 34% to 82% of the storey shear force in X direction, and from 71% to 91% in Y direction, depending on the floor level. Maximum top displacements are 18 and 14 cm in X and Y directions, respectively. The highest interstorey drifts are recorded at the lower floors of about 1.1% and 1.3% in X and Y directions, respectively.

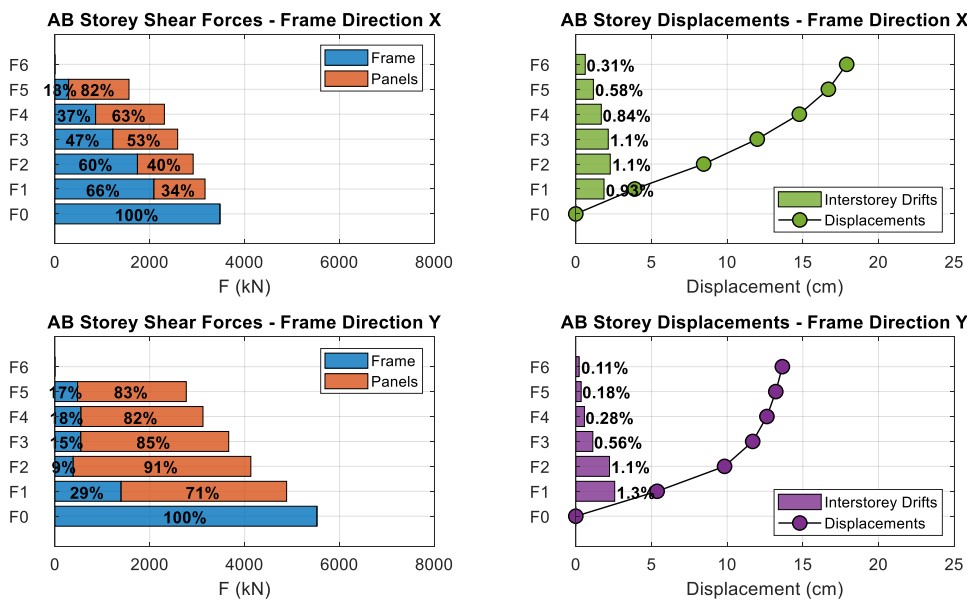

**Figure 10.** As-Built storey shear forces and displacements.

Performance ratios, *pr*, are plotted in Figure 11 charts. The RC beams and columns show a good performance for both brittle and ductile failure modes, with $pr_V$ and $pr_\vartheta$ lower than 1. *pr* values tend to decrease from F1 to F6. It is to notice that steel rebar yielding takes place in both RC beams and columns, as highlighted by the yellow dots plotted in both X and Y directions of chord rotation graphs; however, the chord rotations never reach their ultimate values. Unconfined beam-to-column joints show a good performance in both tensile and compressive terms according to Par. C8.7.2.3.5 of Circ. C.S.LL.PP. 2019 [54]. On the other hand, almost all of the infill panels reach the out-of-plane failure condition, with $pr_M$ values generally higher than 1, leading to a critical scenario in term of masonry

performance. Most of the infill panels also reach the in-plane failure condition, as shown by $pr_d > 1$, except at the higher floor levels, where they are almost completely cracked.

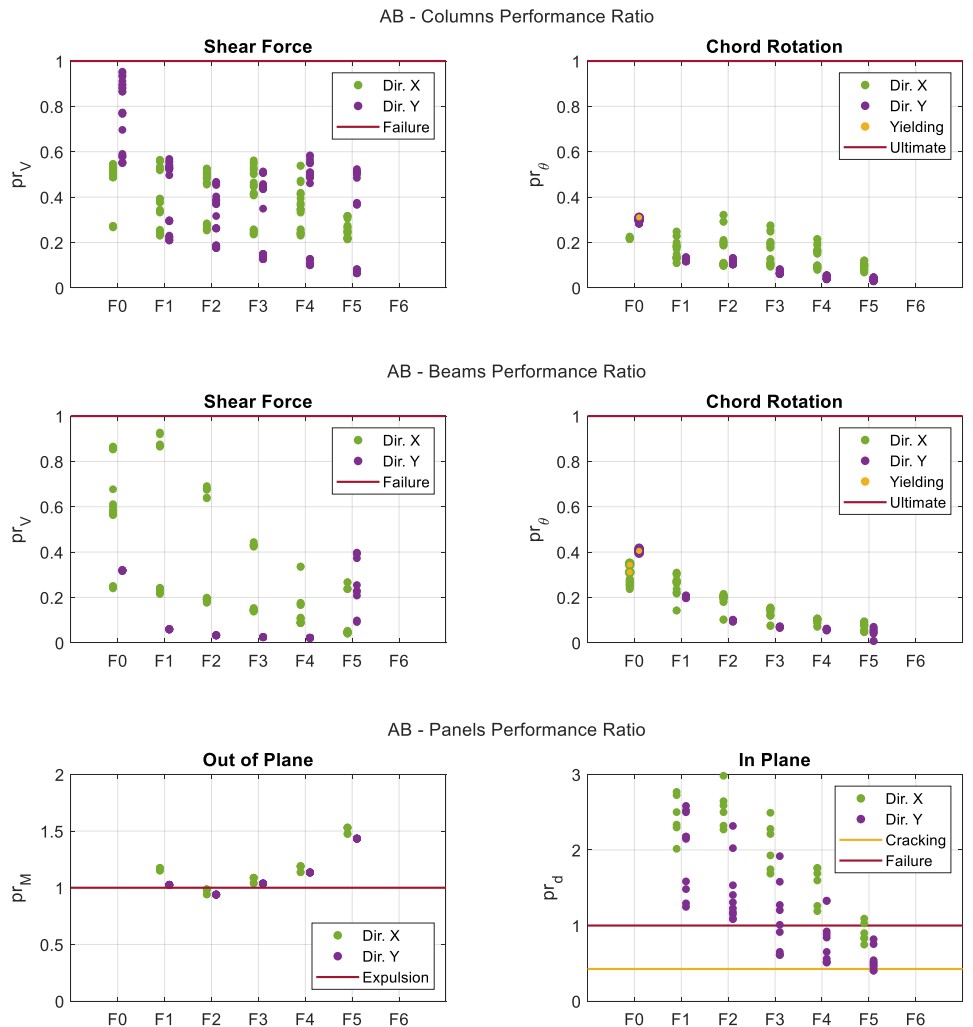

**Figure 11.** As-Built RC elements and infill panels performance ratios.

The AB safety index has been valued as $\alpha = 32\%$, corresponding to a D class of seismic risk, and the EAL has been valued as 1.5% corresponding to a C class of seismic risk, according to the Sismabonus [55]. The overall seismic risk class of the AB configuration has been set to a D level.

## 3. Retrofit Projects Definition and Performance Assessment

Seismic retrofit projects focused on local intervention techniques are herein considered, as AB structural configuration deficiencies mainly involve masonry infill panels. A first option (RP1) consists in strengthening the existing infill panels using FRCM to prevent both in-plane and out-of plane failure modes. A second option (RP2) consists in replacing the existing infill panels with light prefabricated panels, disconnected from the structure in order to avoid structural elements-infill panels interaction.

Seismic retrofit projects that are focused on global intervention techniques are also considered, applying passive control techniques. Accordingly, a third option (RP3) consists in introducing energy dissipation devices, such as FD, along the height of the RC frame and, finally, a fourth option (RP4) consists in introducing base isolation devices, such as LRBs, at the basement level.

The above described four retrofit projects options have been designed and assessed in terms of structural performances and intervention costs, with the aim to compare them under the cost–benefit ratio point of view.

### 3.1. Infill Panels Strengthening (RP1)

In RP1, masonry infill panels are strengthened with FRCM, an innovative composite material that is based on high strength fibers grids embedded into inorganic matrices, such as natural mortars or cements. The application of FRCM at both panel's surfaces can lead to an increment of their in-plane strength up to twice the unreinforced one [56]. Figure 12 shows the comparison between in-plane force-displacement relationships of strengthened and AB infill panels, using the same symbology of Figure 8. In this case study, the effect of FRCM reinforcement has been numerically implemented by considering a double strength value of inelastic struts associated to each panel, and assuming unchanged cracking and ultimate deformations. FRCM application also increases the out-of-plane strength, up to 400% the original one [57], preventing the expulsion failure mode.

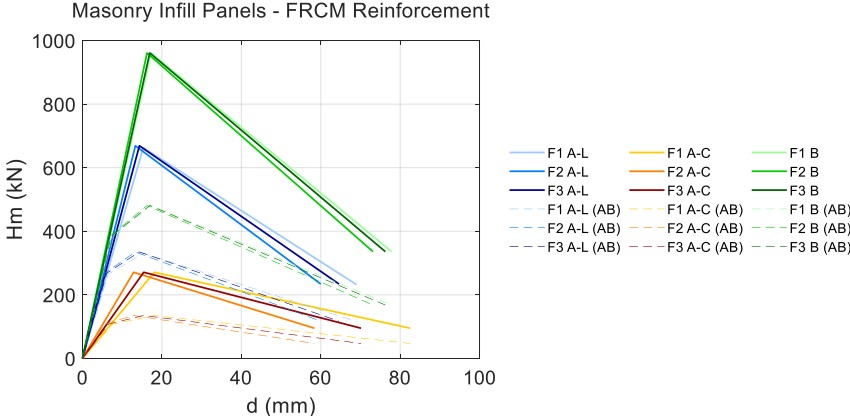

**Figure 12.** Original and strengthened masonry infill panels constitutive models.

Nonlinear time history analysis has been carried out in order to evaluate RP1 structural performance and the obtained results are reported in the following Figures 13 and 14.

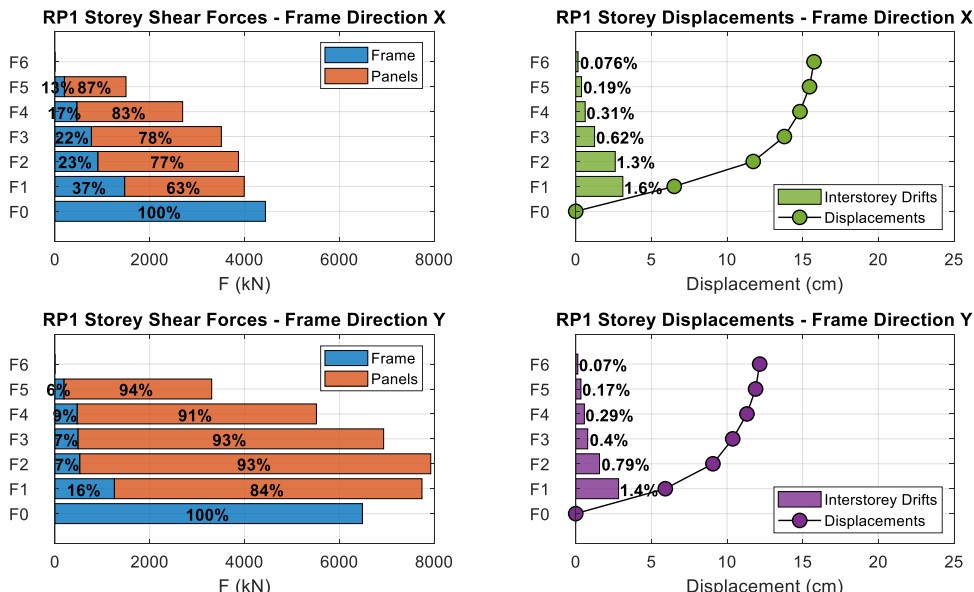

**Figure 13.** RP1 storey shear forces and displacements.

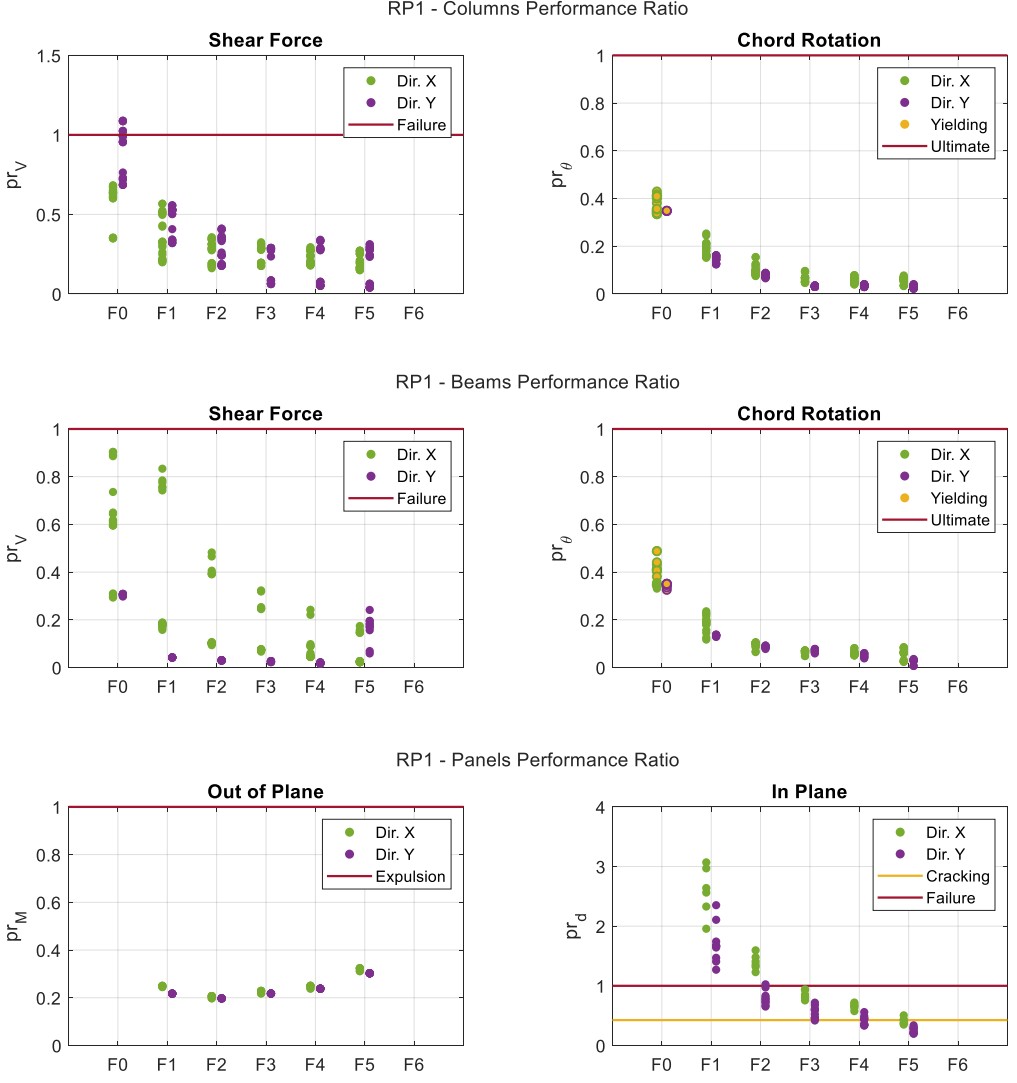

**Figure 14.** RP1 RC elements and infill panels performance ratios.

Storey shear forces and displacements are plotted in Figure 13 charts for both the X and Y direction. Maximum storey shear forces are 4400 kN and 6400 kN in X and Y directions, respectively, which are about 1000 kN higher than the AB configuration in both directions, with an average increment of 21%. Strengthened infill panels absorb from 63% to 87% of the storey shear force in X direction, and from 84% to 94% in Y direction, depending on the floor level, with a shear force average increment of 90% respect to the AB configuration. Maximum top displacements are 16 and 12 cm in X and Y directions, respectively, which are slightly lower than the AB configuration. However, the recorded firsts floors drifts are higher than the AB configuration, leading to values of 1.6% and 1.4% in X and Y directions, respectively.

Performance ratios, *pr*, are plotted in Figure 14 charts. RC beams show a good performance for both brittle and ductile failure modes, with $pr_V$ and $pr_\vartheta$ lower than 1, while RC columns exhibit some shear failures at floor F1, with maximum $pr_V = 1.1$. *pr* values tend to decrease from F1 to F6. It is to notice that steel rebar yielding takes place in both RC beams and columns, as highlighted by the yellow dots that are plotted in both X and Y directions of chord rotation graphs; however, the chord rotations never reach their ultimate values. Unconfined beam-to-column joints show a good performance in both tensile and compressive terms according to Par. C8.7.2.3.5 of Circ. C.S.LL.PP. 2019 [54]. The infill panels show a remarkable out-of plane performance improvement, with $pr_M < 1$. However, in-plane failures still occur at the first two floors, as shown by $pr_d > 1$, with significant damage level at the upper floors.

RP1 safety index has been valued as α = 45%, corresponding to a C class of seismic risk, and the EAL has been valued as 0.93%, corresponding to an A class of seismic risk, according to the Sismabonus [55]. The overall seismic risk class of the RP1 configuration has been set to a C level, highlighting a seismic risk reduction of 1 class with respect to the AB structural configuration.

### 3.2. Infill Panels Replacement (RP2)

In RP2, the masonry infill panels are replaced with light prefabricated panels, set to avoid any interaction with RC frame elements. Typical light prefabricated panels are made by drywall, lighter than masonry panels and settable to allow high interstorey drift values [58,59]. In this case study, the light panels have been numerically implemented as weight only, so that the RP2 structural behavior corresponds to the bare RC frame, without the typical Pilotis behavior.

Nonlinear time history analysis has been carried out in order to evaluate RP2 structural performance and the obtained results are reported in the following Figures 15 and 16.

Storey shear forces and displacements are plotted in Figure 15 charts for both X and Y direction. Maximum storey shear forces are about 3000 kN in both directions, that are lower than the AB configuration of about 500 kN in X direction and 2500 kN in Y direction, with an average decrement of 30%. Maximum top displacements are 25 cm in both directions, which are higher than the AB configuration. Furthermore, the recorded central intersorey drifts are higher than the AB configuration, leading to values of 1.4% in both directions.

Performance ratios, *pr*, are plotted in Figure 16 charts. RC beams and columns show a good performance for both brittle and ductile failure modes, with $pr_V$ and $pr_\vartheta$ lower than 1. The *pr* values tend to increase from F1 to F6. It is to notice that steel rebar yielding takes place in both RC beams and columns, as highlighted by the yellow dots plotted in X direction only of chord rotation graphs; however, the chord rotations never reach their ultimate values. Unconfined beam-to-column joints show a good performance in both tensile and compressive terms according to Par. C8.7.2.3.5 of Circ. C.S.LL.PP. 2019 [54].

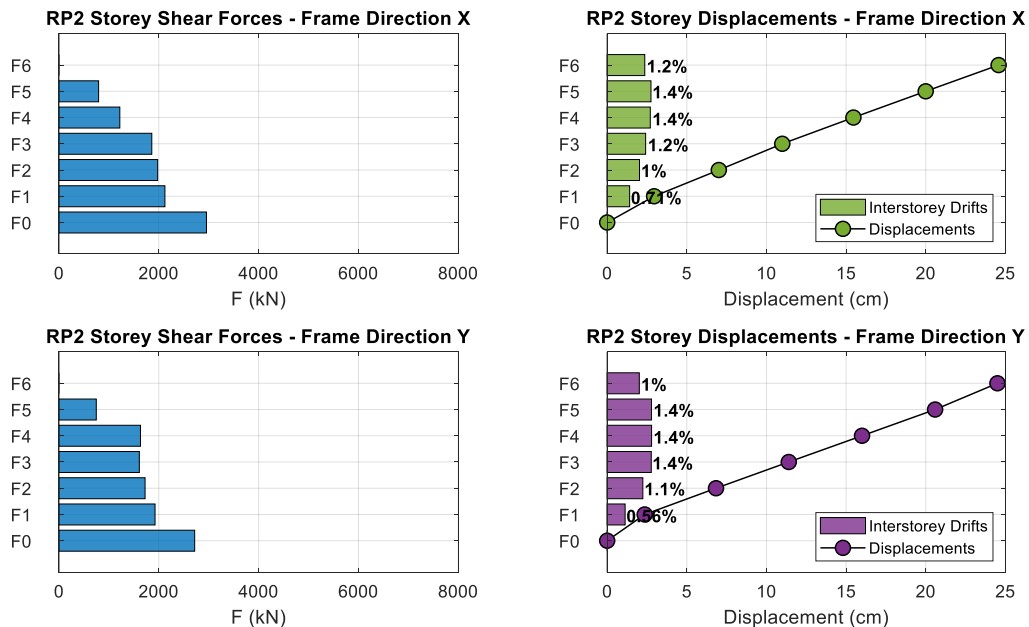

**Figure 15.** RP2 storey shear forces and displacements.

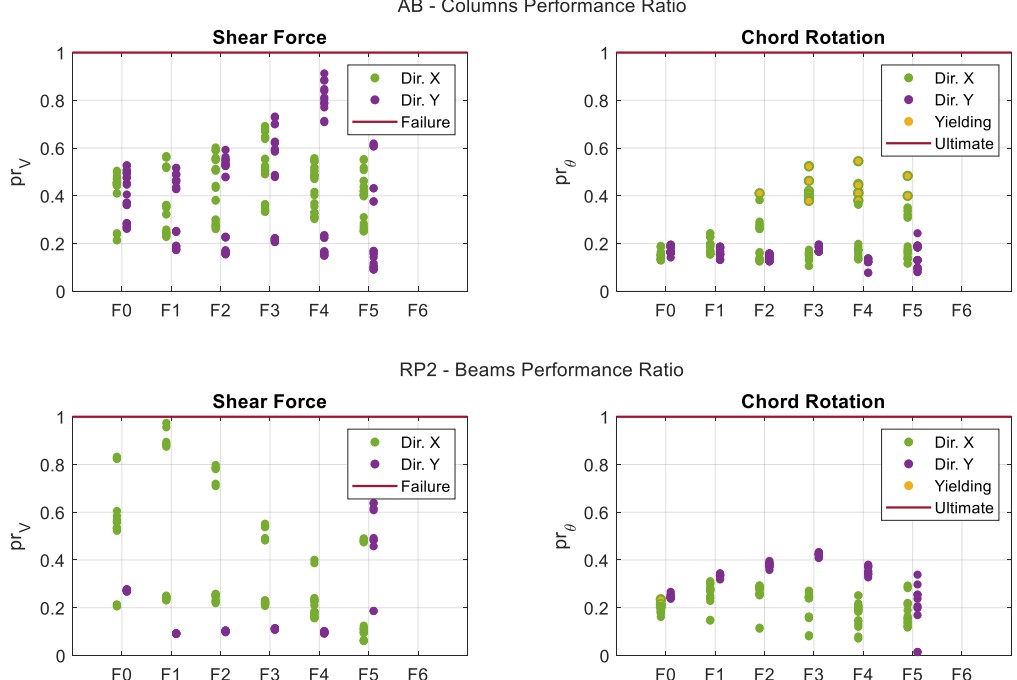

**Figure 16.** RP2 RC elements and infill panels performance ratios.

The RP2 safety index has been valued as $\alpha = 100\%$, corresponding to an A+ class of seismic risk, and the EAL has been valued as 0.46%, corresponding to an A+ class of seismic risk, according to the Sismabonus [55]. The overall seismic risk class of the RP2 configuration has been set to an A+ level, highlighting a seismic risk reduction of four classes with respect to the AB structural configuration.

### 3.3. Energy Dissipation (RP3)

In RP3, energy dissipation is demanded to FDs, distributed within the RC frame respecting its in-plane and in-height regularity. Sized on the AB configuration's interstorey drifts and shear forces, FD devices produced by Quaketek [60] have been selected. Figure 17 shows the implemented FDs constitutive model, where an activation force of about 800 kN and a maximum allowable displacement of 94 mm can be observed. Thanks to FDs application, four bracing frames are added to the AB RC frame in both X and Y directions, as schematically represented in Figure 18.

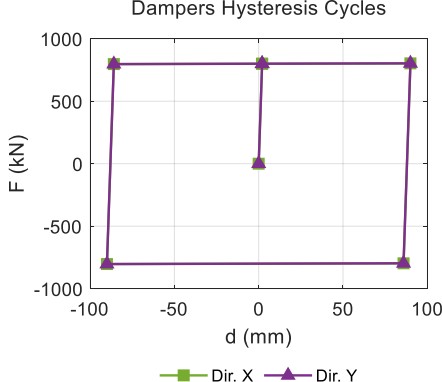

**Figure 17.** Selected Friction Dampers (FD) Hysteresis Cycle.

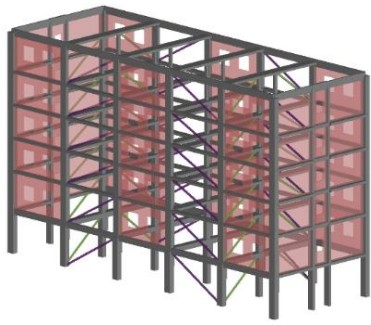

**Figure 18.** RP3 principal structure.

Nonlinear time history analysis has been carried out in order to evaluate RP1 structural performance and the obtained results are reported in the following Figures 19 and 20.

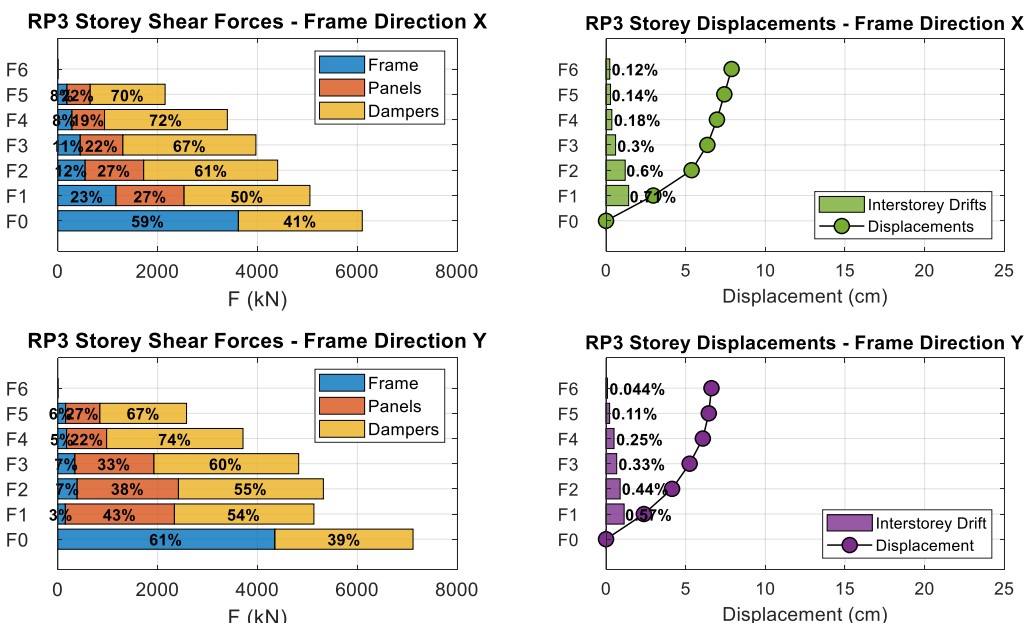

**Figure 19.** RP3 storey shear forces and displacements.

Storey shear forces and displacements are plotted in Figure 19 charts for both X and Y direction. Maximum storey shear forces are 6100 kN and 7100 kN in X and Y directions, respectively, which are about 2000 kN higher than the AB configuration in both directions, with an average increment of 52%. The charts show that infill panels absorb from 19% to 22% of the storey shear force in X direction, and from 22% to 43% in Y direction, depending on the floor level. The FD devices absorb from 41% to 72% of the storey shear force in X direction, and from 39% to 74% in Y direction, depending on the floor level. Infill panels and frame columns register a shear force average decrement of 50% respect to the AB configuration, except F1 frame columns that register a shear force average decrement of 10% respect to the AB configuration. Maximum top displacements are 8 and 7 cm in X and Y directions, respectively, which are considerably lower values with respect to the AB configuration. F1 drifts are lower than the AB configuration, leading to values of 0.71% and 0.57% in X and Y directions, respectively.

Performance ratios, *pr*, are plotted in Figure 20 charts. RC beams and columns show a good performance for both brittle and ductile failure modes, with $pr_V$ and $pr_\vartheta$ lower than 1. The *pr* values tend to decrease from F1 to F6. It is to notice that steel rebar yielding takes place in RC beams only, as highlighted by the yellow dots plotted in X direction only of chord rotation graphs; however, the chord rotations never reach their ultimate values. Most of the infill panels reach the out-of-plane failure condition with $pr_M$ higher than 1, while they reach in-plane failure condition at F1 with $pr_d > 1$, and

limited damage level at the upper floors. Unconfined beam-to-column joints show a good performance in both tensile and compressive terms according to Par. C8.7.2.3.5 of Circ. C.S.LL.PP. 2019 [54].

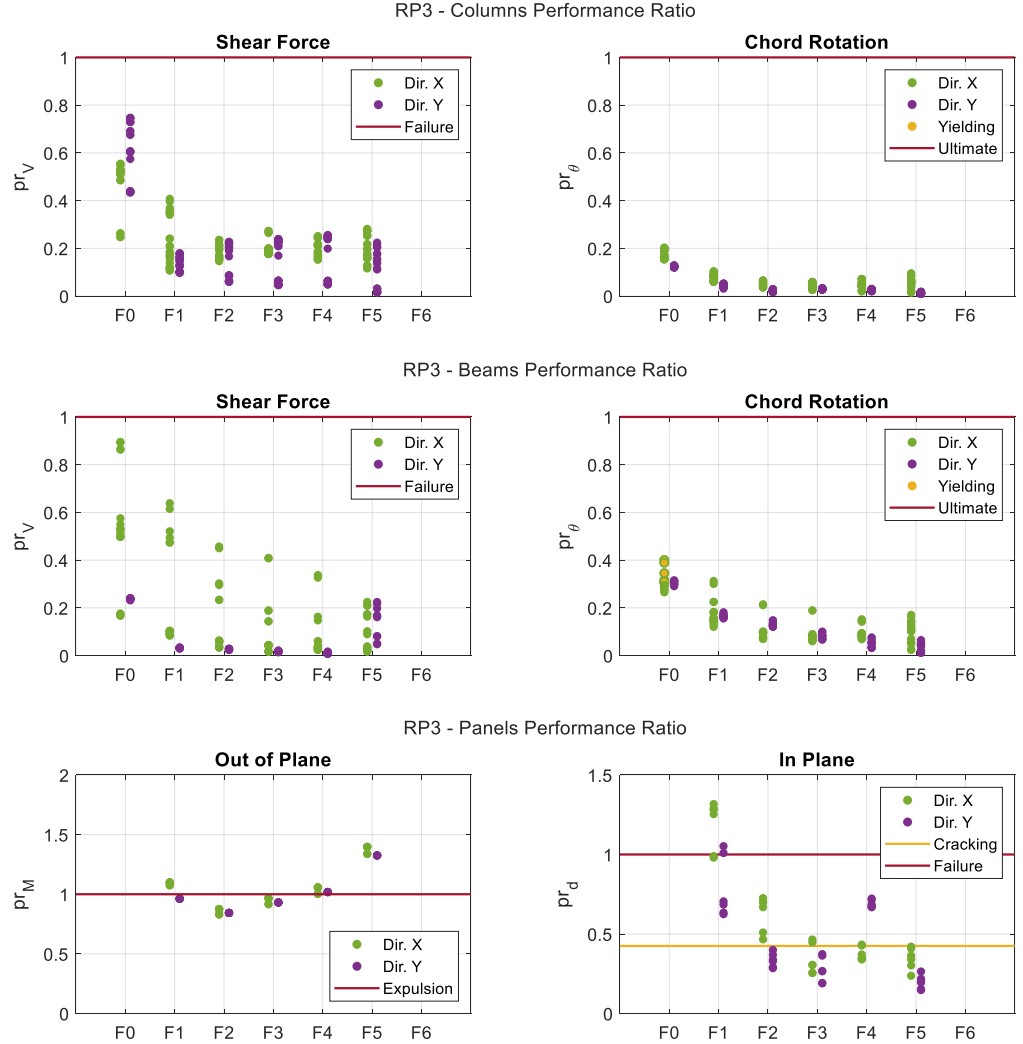

**Figure 20.** RP3 RC elements and infill panels performance ratios.

RP3 safety index has been valued as α = 62%, corresponding to a B class of seismic risk, and the EAL has been valued as 0.65%, corresponding to an A class of seismic risk, according to the Sismabonus [55]. The overall seismic risk class of the RP3 configuration has been set to a B level, highlighting a seismic risk reduction of 2 class with respect to the AB structural configuration.

### 3.4. Base Isolation (RP4)

In RP3, the base isolation technique is applied by using LRBs produced by FipIndustriale [61]. Figure 21 shows the implemented LRBs constitutive model, where an activation force of about 200 kN, a maximum force of 400 kN and a maximum allowable displacement of 333 mm can be observed. Sliding Bearings (SB) that are produced by FipIndustriale [62] are also used at the basement level, together with LRBs, in order to obtain a regular in-plane behavior. SB activation force and maximum allowable displacement are about 200 kN and 50 mm, respectively. Figure 22 shows the overall base isolated structure, where LRBs and SBs, located at the base, are represented in blue and red colors, respectively. It can be observed that a new rigid RC diaphragm must be created at the basement level before setting LRBs and SBs.

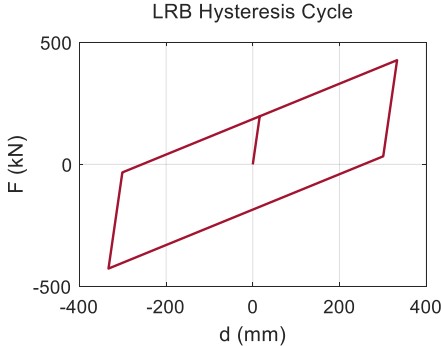

**Figure 21.** Selected LRB hysteresis cycle.

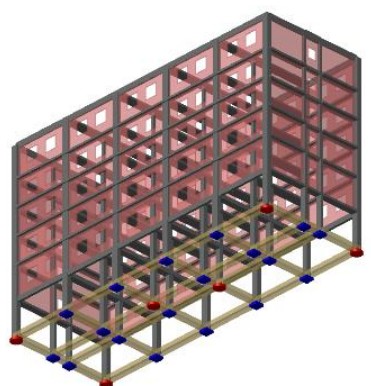

**Figure 22.** RP4 principal structure.

Nonlinear time history analysis has been carried out to evaluate RP4 structural performance and the obtained results are reported in the following Figures 23 and 24.

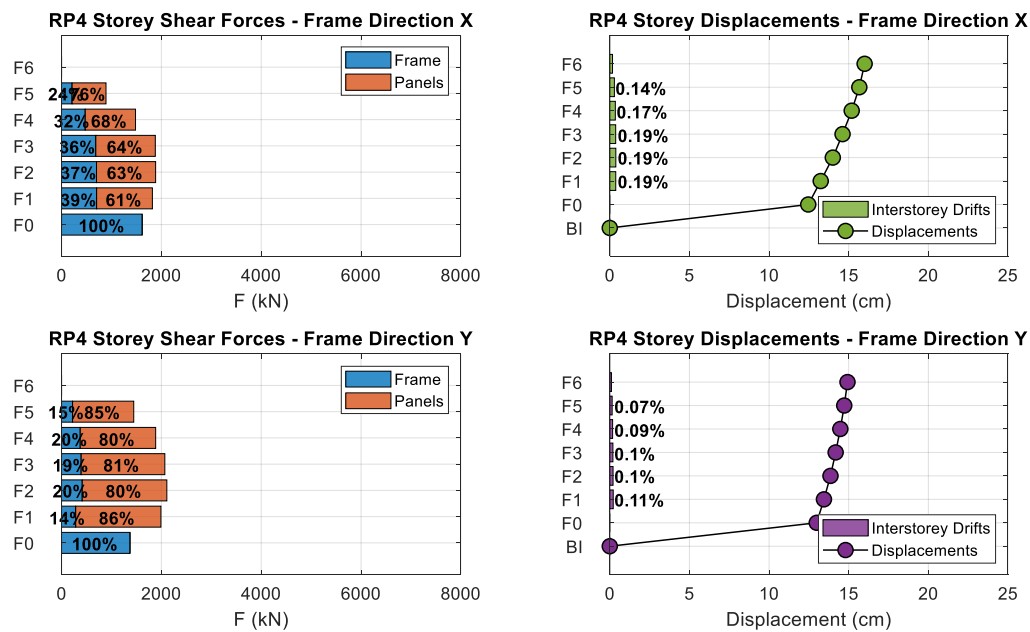

**Figure 23.** RP4 storey shear forces and displacements.

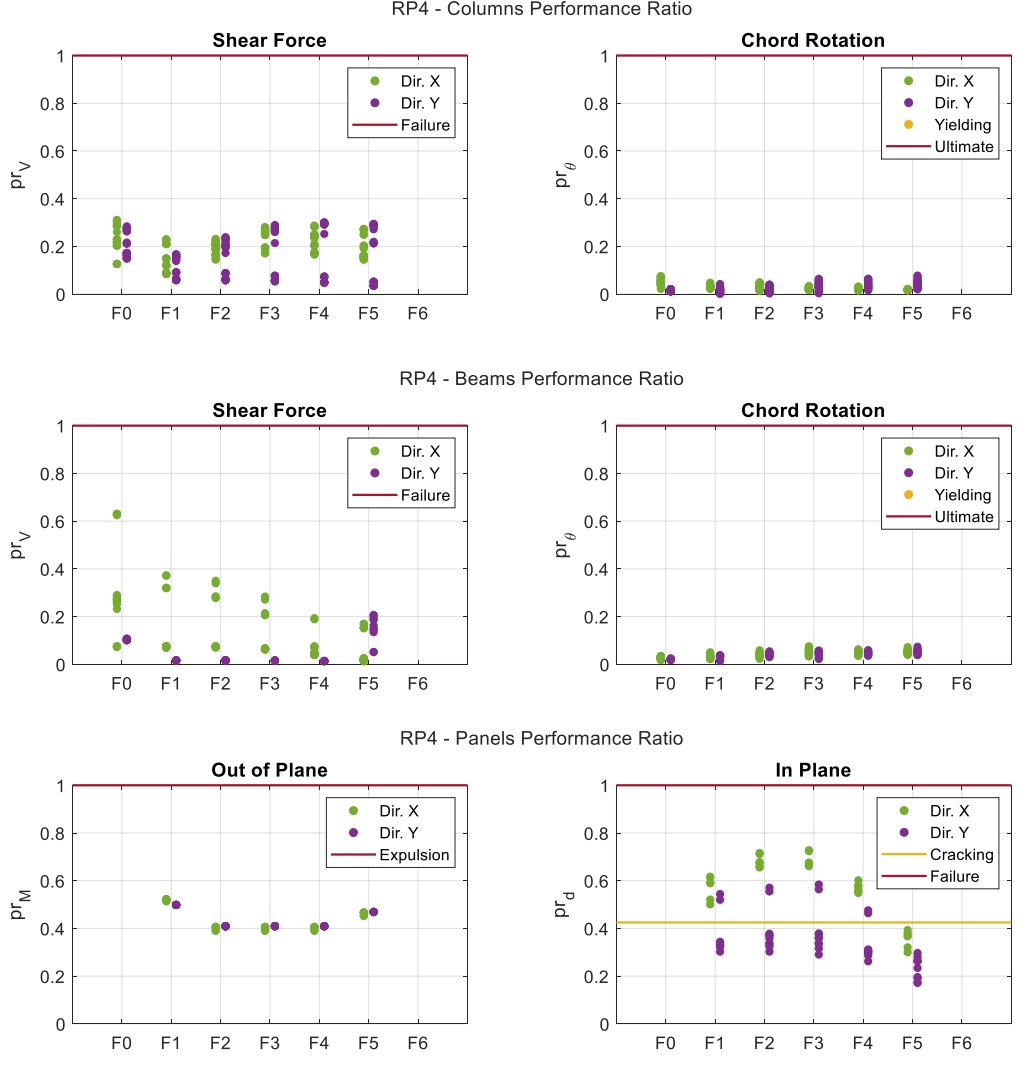

**Figure 24.** RP4 RC elements and infill panels performance ratios.

Storey shear forces and displacements are plotted in Figure 23 charts for both the X and Y direction. Maximum storey shear forces are about 1500 kN in both directions, which are lower than the AB configuration of about 2000 kN in X direction and 4000 kN in Y direction, with an average decrement of 65%.Infill panels absorb from 61% to 76% of the storey shear force in X direction, and from 80% to 86% in Y direction, depending on the floor level, with a shear force average decrement of 90% respect to the AB configuration. The infill panels and frame columns register a shear force average decrement of 65% with respect to the AB configuration and of 62% with respect to the RP3 configuration. Maximum top displacements are 16 and 15 cm in X and Y directions, respectively, which are higher than the AB configuration. However, recorded F1 drifts are remarkably lower than the AB configuration, leading to values of 0.19% and 0.11% in X and Y directions, respectively.

Performance ratios, *pr*, are plotted in Figure 24 charts. RC beams and columns show a good performance for both brittle and ductile failure modes, with $pr_V$ and $pr_\vartheta$ lower than 1. *pr* values tend to decrease from F1 to F6. It is to notice that steel rebar yielding does not take place in both RC beams and columns. Infill panels show remarkable out-of plane and in-plane performances improvement, with $pr_M < 1$ and $pr_d < 1$, with a moderate in-plane damage level only. Unconfined beam-to-column joints show a good performance in both tensile and compressive terms according to Par. C8.7.2.3.5 of Circ. C.S.LL.PP. 2019 [54].

The RP4 safety index has been valued as α = 100%, corresponding to a A+ class of seismic risk, and the EAL has been valued as 0.45%, corresponding to a A+ class of seismic risk, according to the

Sismabonus [15]. The overall seismic risk class of the RP4 configuration has been set to a A+ level, highlighting a seismic risk reduction of 4 class with respect to the AB structural configuration.

## 4. Performance Comparison of Retrofit Projects

In order to thoroughly compare RP1, RP2, RP3, and RP4 to AB structures in terms of structural behavior, their performance has been plotted on the Acceleration-Displacement Response Spectrum (ADRS) graph of Figure 25 for the SD Limit State of the considered site according to NTC 2018 [21].

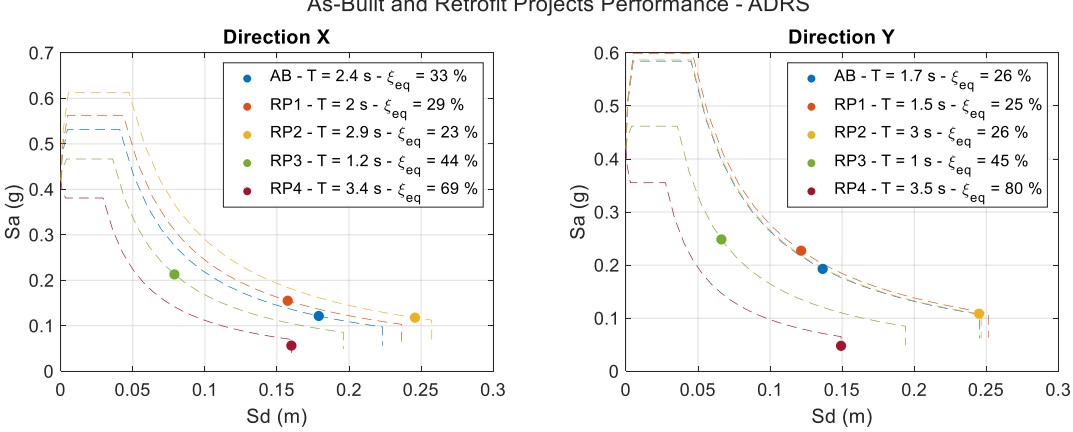

**Figure 25.** As-Built (AB) and RPs performance Acceleration-Displacement Response Spectrum (ADRS) in X and Y directions.

RP1 and RP2 are based on local interventions. In terms of energy dissipation rate, they provide an average equivalent damping in X and Y directions, $\xi_{eq}$, of about 26% very similar to the 29% of the AB configuration. RP1 fundamental periods, T, in X and Y directions (2 s and 1.5 s, respectively) are very similar to the AB configuration (2.4 s and 1.7 s, respectively), while RP2 fundamental periods (2.9 s and 3 s, respectively) are higher than the AB configuration due to the lack of infill panels-RC frame interaction. The RP1 and AB solutions have similar spectral accelerations (about 0.15 g and 0.2 g in X and Y directions, respectively) and displacements (about 17 cm and 13 cm in X and Y directions, respectively) while RP2 has lower spectral acceleration (about 0.1 g in both X and Y directions) and higher displacement (about 25 cm in both X and Y directions).

RP3 and RP4 are based on global interventions. In terms of energy dissipation rate, they provide an average equivalent damping in X and Y directions, $\xi_{eq}$, of about 45% and 75%, respectively, much higher than the 29% of the AB configuration. RP3 fundamental periods, T, in X and Y directions (on average 1.1 s) are much lower than the AB configuration (2.4 s and 1.7 s, respectively), while RP4 fundamental periods (in average 3.4 s) are much higher than the AB configuration. RP3 has higher spectral acceleration (in average 0.23 g in X and Y directions) and lower displacement (in average 7 cm in both X and Y directions) when compared with the AB configuration. On the contrary, RP4 has lower spectral acceleration (in average 0.05 g in X and Y directions) and higher displacement (on average 25 cm in both X and Y directions) if compared with the AB configuration.

It is interesting to underline that RP2 and RP4 equalize the structural performance in X and Y directions, decrease the seismic actions more efficiently than RP1 and RP3, and provide a safety level as high as the expected for a new construction.

A thorough structural performance evaluation must account both safety and damage issues. The proposed retrofit projects can have similar safety indexes, but much different expected damage levels, as outlined by the performance assessment presented in Section 3.

In order to perform a full comparison, damage indexes for frame elements, $\rho_\theta$, and infill panels, $\rho_d$, have been set as ratio between demand and capacity of the deformation excursions in the inelastic field,

and they have been expressed as function of ductility demands $\mu_{\theta,D}, \mu_{d,D}$, and capacities $\mu_{\theta,C}, \mu_{d,C}$ using the following Equations (14) and (15):

$$\rho_\theta = \frac{\theta - \theta_y}{\theta_u - \theta_y} = \frac{\mu_{\theta,D} - 1}{\mu_{\theta,C} - 1} \tag{14}$$

$$\rho_d = \frac{d - d_{cr}}{d_u - d_{cr}} = \frac{\mu_{d,D} - 1}{\mu_{d,C} - 1} \tag{15}$$

where:

- $\theta$ and $d$ are the frame elements chord rotation and panels displacement demand, respectively;
- $\theta_y$ and $d_{cr}$ are the frame elements chord rotation at yielding and panels displacement at cracking, respectively; and,
- $\theta_u$ and $d_u$ are the frame elements chord rotation and panels displacement capacity, respectively.

Global damage indexes for frame elements, $DI_\theta$, and infill panels, $DI_d$, have also been defined as ratio between summations, over the *i*-th structural elements, of demand and capacity of the deformation excursions in the inelastic field, as given by (16) and (17) correspondingly. Global damage indexes $DI_\theta$ and $DI_d$ have been set to range between 0 and 1, where 0 means no damage and 1 means the ultimate failure of all the structural elements. Global damage indexes have been introduced in order to measure how much the damaged elements affect the overall structure.

$$DI_\theta = \frac{\sum_i (\theta - \theta_y)_i}{\sum_i (\theta_u - \theta_y)_i} = \frac{\sum_i (\mu_{\theta,D} - 1)_i}{\sum_i (\mu_{\theta,C} - 1)_i} \tag{16}$$

$$DI_d = \frac{\sum_i (d - d_{cr})_i}{\sum_i (d_f - d_{cr})_i} = \frac{\sum_i (\mu_{d,D} - 1)_i}{\sum_i (\mu_{d,C} - 1)_i} \tag{17}$$

In addition, the total number of damaged frame elements, $DE_{tot,\theta}$, and infill panels, $DE_{tot,d}$, have been set as ratio between the number of frame elements and infill panels with damage index $\rho_\theta$, $\rho_d > 0$, and the total number of frame elements and infill panels, respectively. Finally, the total number of failed frame elements, $FE_{tot,\theta}$, and infill panels, $FE_{tot,d}$, have been set as the ratio between the number of frame elements and infill panels with damage index $\rho_\theta$, $\rho_d = 1$, and the total number of frame elements and infill panels, respectively.

Table 7 reports frame elements and infill panels DI, $DE_{tot}$ and $FE_{tot}$ in percentage.

**Table 7.** DI, $DE_{tot}$, and $FE_{tot}$ for AB and RPs configurations.

| | **Frame Elements** | | | | | **Infill Panels** | | | | |
|---|---|---|---|---|---|---|---|---|---|---|
| | **AB** | **RP1** | **RP2** | **RP3** | **RP4** | **AB** | **RP1** | **RP2** | **RP3** | **RP4** |
| $DI_{\theta,d}$ (%) | 0.44 | 0.99 | 0.65 | 0.11 | 0 | 87.37 | 51.98 | 0 | 27.84 | 12.53 |
| $DE_{tot,\theta,d}$ (%) | 6.58 | 10.31 | 6.80 | 1.32 | 0 | 75.00 | 67.50 | 0 | 36.67 | 40.00 |
| $FE_{tot,\theta,d}$ (%) | 0 | 0 | 0 | 0 | 0 | 43.75 | 21.25 | 0 | 3.33 | 0 |

Frame elements $DI_\theta$ are always lower than 1%; however, the $DI_\theta$ values of RP1 and RP2 are higher while RP3 and RP4 are lower than the AB configuration ($DI_\theta = 0.44\%$). The better performance is given by RP4 with a $DI_\theta = 0\%$; the worst performance is given by RP1 with a $DI_\theta = 0.99\%$. Infill panels $DI_d$ strongly depend on the retrofit configuration; $DI_d$ values of all the RPs are lower than the AB configuration ($DI_d = 87.37\%$). The better performance is given by RP2 with a $DI_d = 0\%$; the worst performance is given by RP1 with a $DI_\theta = 51.28\%$.

Frame elements $DE_{tot,\theta}$ are always lower than 10%; however, $DE_{tot,\theta}$ values of RP1 and RP2 are higher, while RP3 and RP4 are lower than the AB configuration ($DE_{tot,\theta} = 6.58\%$). The better

performance is given by RP4 with a $DE_{tot,\theta} = 0\%$; the worst performance is given by RP1 with a $DE_{tot,\theta} = 10.31\%$. The infill panels $DE_{tot,d}$ strongly depend on the retrofit configuration; $DE_{tot,d}$ values of all the RPs are lower than the AB configuration ($DE_{tot,d} = 75\%$). The better performance is given by RP2 with a $DE_{tot,d} = 0\%$; the worst performance is given by RP1 with a $DE_{tot,d} = 67.50\%$.

Frame elements $FE_{tot,\theta} = 0\%$ for all of the considered structural configuration. Infill panels $FE_{tot,d}$ strongly depend on the retrofit configuration; $FE_{tot,d}$ values of all the RPs are lower than the AB configuration ($FE_{tot,d} = 43.75\%$). The better performance is given by RP2 and RP4 with a $FE_{tot,d} = 0\%$; the worst performance is given by RP1 with a $FE_{tot,d} = 21.25\%$.

It is interesting to highlight that RP2 and RP4 result to be the more efficient RP configurations, providing the better performance for both infill panels and frame elements.

The number of frame elements, $DE_\theta$, and infill panels, $DE_d$, which have reached a certain level of damage, $\rho_\theta$ and $\rho_d$, respectively, have also been computed and expressed in percent respect the number of total elements. $DE_\theta$ and $DE_d$ are evaluated in distributed form, i.e., as total damaged elements' percentage for each $\rho_\theta, \rho_d$, interval and, in cumulative form, i.e., as total damaged elements' percentage up to the $\rho_\theta, \rho_d$, interval. In Figure 26, $DE_\theta$ and $DE_d$ in both distributed and cumulative forms are plotted vs. $\rho_\theta$ and $\rho_d$, respectively, with $\rho_\theta, \rho_d$, intervals of 0.1.

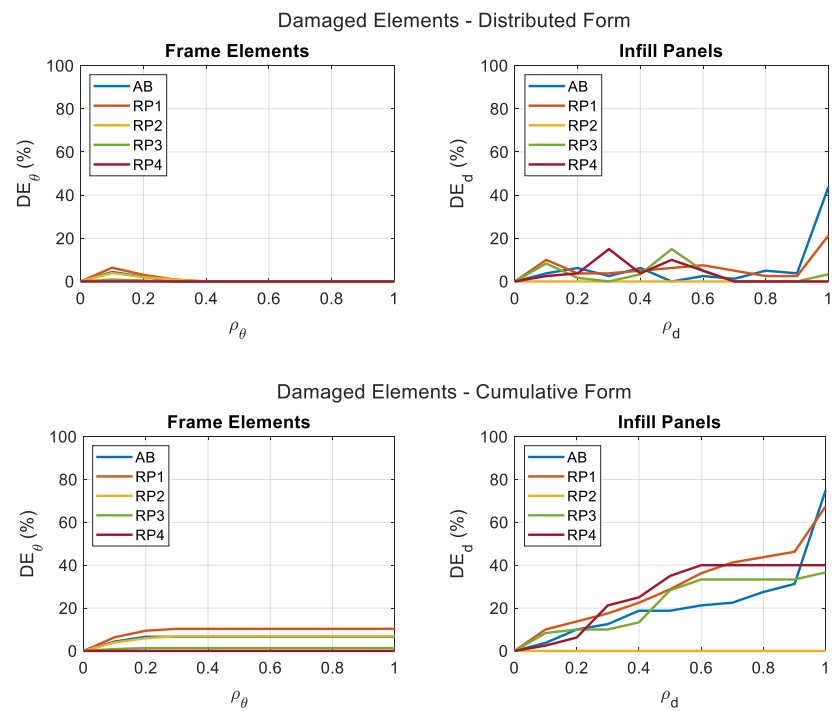

**Figure 26.** DE in distributed and cumulative forms for AB and RPs configurations.

For frame elements, $DE_\theta$ maximum value is lower than 10% for a $\rho_\theta$ of about 0.1 for all of the RPs. The number of elements that reach higher values of $\rho_\theta$ are progressively lower, as confirmed by the cumulative plot of $DE_\theta$ that is set constant for $\rho_\theta > 0.3$. The damage level of frame elements is low and involves a limited percentage of elements (about 10%) for all of the proposed RPs configuration. RP4 configuration still highlights the better performance due to damage lacking.

For infill panels, $DE_d$ maximum value is about 40% for a $\rho_d$ of about 1 for the AB configuration, 20% for the RP1 and lower than 5% for the other RPs. The cumulative plot of $DE_d$ increases linearly with $\rho_d$, due to an almost uniform damage distribution for all of the RPs. The damage level of infill panels is generally high and involve a significant percentage of elements (between 40 and 80%), depending on the proposed RPs configuration. RP2 configuration highlights the better performance due to damage lacking; however, RP4 exhibits a good performance in term of infills damage, reaching a maximum $\rho_d = 0.6$ for less than 10% of elements, without any failure occurrence.

A cost analysis has been performed for the considered RPs in order to complete the retrofit projects comparison. Retrofit total costs are classified in direct and indirect costs. Direct costs include the structural and related non-structural work costs, estimated using Italian medium prices [63]. Indirect costs take in account the occupants dislocation costs during the rehabilitation works, estimated using Italian month-rent medium costs for a 100 m$^2$ apartment and including the moving costs. Dislocation time has been estimated based on a single work-month for each building floor. Eventual repair costs have also been considered for future post-earthquake rehabilitation interventions, estimated using Italian medium prices [63]. The repair costs include the strengthening of yielded frame elements using FRP, the demolition and reconstruction of collapsed masonry infill panels and the restoration of cracked panels, but they do not include the costs of eventual displaced occupants, since minor damage levels are expected.

Retrofit total costs and eventual repair costs that are evaluated in Euros are collected in Table 8 together with the achieved risk levels for the considered RPs and the expected seismic risk reduction with respect to the AB structural configuration. Retrofit total costs and eventual repair costs in Euros and in Euros per unit surface are also graphically compared in Figure 27.

**Table 8.** Achieved seismic risk level, retrofit, and repair costs for different RPs.

| | RP1 | RP2 | RP3 | RP4 |
|---|---|---|---|---|
| Seismic Risk Level | C (+1) | A+ (+4) | B (+2) | A+ (+4) |
| Direct Costs | 450,000€ 201 €/m$^2$ | 420,000€ 187 €/m$^2$ | 880,000€ 393 €/m$^2$ | 300,000€ 134 €/m$^2$ |
| Indirect Costs | 90,000€ 40 €/m$^2$ | 90,000€ 40 €/m$^2$ | 90,000€ 40 €/m$^2$ | 0€ 0 €/m$^2$ |
| Total Costs | 540,000€ 241 €/m$^2$ | 510,000€ 228 €/m$^2$ | 970,000€ 433 €/m$^2$ | 300,000€ 134 €/m$^2$ |
| Eventual Repair Costs | 200,000€ 89 €/m$^2$ | 70,000€ 31 €/m$^2$ | 170,000€ 76 €/m$^2$ | 30,000€ 13 €/m$^2$ |

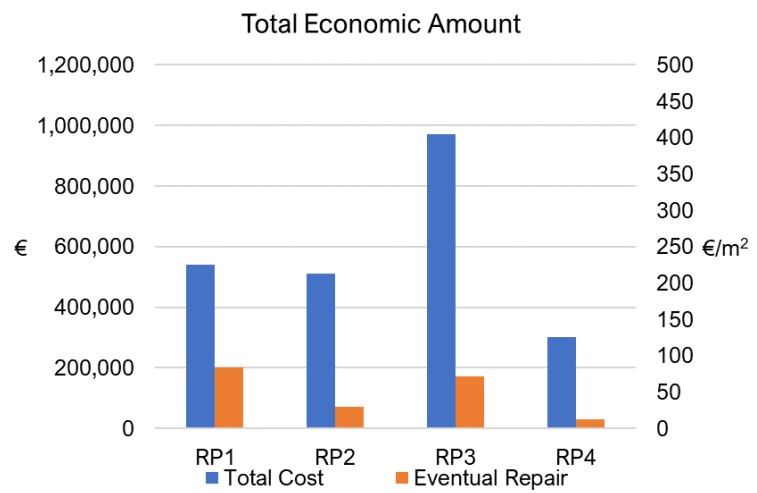

**Figure 27.** Total and per unit surface costs for different RPs.

Based on the obtained results, RP4 is by far the most efficient intervention technique among the considered RPs, achieving the highest safety levels with the lowest total costs. Differently from the other RPs, RP4 provides very low storey accelerations during a major earthquake and the occupants' danger perception can be very reduced. Furthermore, RP4 exhibits lower eventual post-earthquake repair costs, due to the lack of post-earthquake structural damage, while slightly damaged infill panels

can be easily and inexpensively restored. Finally, RP4 execution does not interfere with the building occupants' life, and current activities are not interrupted during the whole duration of rehabilitation works. RP2 also reveals to be a very promising retrofit strategy, achieving the highest safety level with limited damage and relatively low costs. However, due to the high impact of rehabilitation works on the occupants' lifestyle, it is mainly suggested for the retrofit of unoccupied buildings.

## 5. Conclusions

In this study, four different retrofit projects have been developed and compared for the seismic rehabilitation of an existing Pilotis RC frame, designed for gravity loads only and located in a high seismic risk region of Italy. Advanced retrofit techniques have been applied both at the local and global structural level. Two retrofit projects have been defined at the local level by strengthening the original masonry infilled panels with FRCM (RP1) and alternatively replacing the original infilled panels with not interacting and light prefabricated panels (RP2). Two more retrofit projects have been defined at the global level by improving the overall structural capacity by means of FDs (RP3) and LRBs (RP4) applications. Nonlinear time history analysis has been carried out for as-built (AB) and retrofit (RPs) structural configurations in order to assess the structural performance according to Eurocode 8 [22] and NTC 2018 [21].

The main remarks are collected below:

- In RP1 configuration, in-plane and out-of-plane stiffness and strength of infilled masonry panels are much higher than in the AB configuration, due to the FRCM application. The overall structural stiffness and base shear force are also increased and shear failure and yielding occur in some first-floor RC beams and columns, without reaching the ultimate chord rotation. Thanks to RP1, a Seismic Risk Class type C is assessed according to Sismabonus [55]. RP1 provides an average equivalent damping $\xi_{eq}$ in X and Y directions of about 27%, very similar to the 29% of the AB configuration. RC frame elements and masonry infilled panels result in being slightly and moderately damaged with a $DI_\theta$ of 0.99% and a $DI_d$ of 52%, respectively. Structural damage affects the 10.31% of RC frame elements, without failure occurrence, and the 67.50% of infill panels, with failure occurrence of 21.25%. RP1 realization cost is about 241 €/m$^2$ with an eventual repair cost of 89 €/m$^2$ after a major earthquake. Furthermore, occupants are accounted to leave the building during both the rehabilitation and repair interventions.

- In RP2 configuration, original infilled masonry panels are removed and replaced by non-interacting light prefabricated panels, so that the activated seismic response corresponds to the bare frame seismic response. The overall structural stiffness is substantially reduced due to the abolition of the Pilotis frame behavior, leading to lower base shear forces and higher inter-storey drifts and top displacements. RC columns yielding occurs at different storey heights, without reaching the ultimate chord rotation. Thanks to RP2, a Risk Class A+ is assessed according to Sismabonus [55]. RP2 provides an average equivalent damping $\xi_{eq}$ in X and Y directions of approximately 25%, very similar to the 29% of the AB configuration. RC frame elements and masonry infilled panels result slightly and no damaged with a $DI_\theta$ of 0.65% and a $DI_d$ of 0%, respectively. Structural damage affects the 6.80% of RC frame elements, without failure occurrence. RP2 realization cost is about 228 €/m$^2$ with an eventual repair cost of 31 €/m$^2$ after a major earthquake. Furthermore, occupants are accounted to leave the building during both the rehabilitation and repair interventions.

- In RP3 configuration, the global performance under seismic actions is improved by using energy dissipation that is supplied by FDs system. The overall structural stiffness is increased, leading to a substantial reduction of inter-storey drifts and top displacements together with a meaningful increase of base shear forces. However, the frame demand is decreased due to FDs system efficacy and RC columns yielding occurs in very few elements, without reaching the ultimate chord rotation. Thanks to RP3, a Risk Class B is assessed according to Sismabonus [55]. RP3 provides an average equivalent damping $\xi_{eq}$ in X and Y directions of about 45%, higher than the 29% of the AB configuration. RC frame elements and masonry infilled panels result in slightly and moderately

damaged with a $DI_\theta$ of 0,11% and a $DI_d$ of 28%, respectively. Structural damage affects the 1.32% of RC frame elements, without failure occurrence, and the 36.67% of infill panels, with failure occurrence of 3.33%. RP3 realization cost is about 433 €/m$^2$ with an eventual repair cost of 76 €/m$^2$ after a major earthquake. Furthermore, occupants are accounted to leave the building during both the rehabilitation and repair interventions.

- In RP4 configuration, the global performance under seismic action is improved using base isolation that is supplied by LRBs system. The main oscillation period is conveniently increased, leading to a meaningful reduction of base shear forces and inter-storey drifts even though with increased top displacements. Thanks to RP4, a Risk Class A+ is assessed according to Sismabonus [55]. RP4 provides an average equivalent damping $\xi_{eq}$ in X and Y directions of about 75%, much higher than the 29% of the AB configuration. RC frame elements results without damage, and masonry infilled panels result in being slightly damaged with a $DI_\theta$ of 0% and a $DI_d$ of 13%, respectively. Structural damage affects the 40.00% of infill panels, without failure occurrence. RP4 realization cost is about 134 €/m$^2$ with an eventual repair cost of 13 €/m$^2$ after a major earthquake. Differently by previous cases, occupants are not accounted to leave the building during both the rehabilitation and repair interventions.

The base isolation technique, based on LRBs application, has provided the better overall structural performance with much higher equivalent damping and much lower associated costs than the other considered rehabilitation techniques, confirming to be a very effective and competitive retrofit technique for Pilotis RC frames rehabilitation. Additionally, occupants' activities are not interrupted during the whole duration of rehabilitation works. However, base isolation technique is not always permitted, as in the presence of nearby buildings or soft soils, so the other retrofit configurations can still be very useful as design tools for professionals.

Energy dissipation technique, based on FDs application, has also proved to be a sound solution, but with high associated costs despite the high equivalent damping provided. The FRCM strengthening technique is suggested as complementary technique only, in case base isolation or energy dissipation techniques need further interventions due to a highly vulnerable as-built structure. Finally, the replacement of original masonry infill panels with non-interacting light prefabricated panels has also proved to be a very promising retrofit strategy for Pilotis RC frames. However, due to the high impact of rehabilitation works on the occupants' lifestyle, it is suggested mainly for the retrofit of unoccupied buildings.

It is to recall that the considered structure was originally designed to host essential functions, with high quality RC elements detailing if compared to the standard detailing of the construction time. The considered rehabilitation techniques can lead to different results when applied to RC frames with poor detailing or different structural typologies. This work also emphasizes that non-linear numerical analysis provides a more accurate assessment of the existing structures performance, fully taking into account the available ductility and capacity of the structural elements, including infill panels, and helping professionals to identify the better rehabilitation technique according to different requirements.

**Author Contributions:** Conceptualization, M.Z. and A.A.; methodology, E.G., M.Z. and A.A.; software, E.G.; validation, E.G., M.Z. and A.A.; formal analysis, E.G.; investigation, E.G., M.Z. and A.A.; resources, M.Z. and A.A.; data curation, E.G.; writing—original draft preparation, E.G.; writing—review and editing, M.Z. and A.A.; visualization, E.G.; supervision, M.Z. and A.A.; project administration, M.Z. and A.A.; funding acquisition, A.A. All authors have read and agreed to the published version of the manuscript.

**Funding:** The authors wish to acknowledge the support provided by FAR funding from the University of Ferrara, 2019–2020, "In-plane shear capacity of clay masonry walls strengthened with textile reinforced mortars (TRM)".

**Conflicts of Interest:** The authors declare no conflict of interest.

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
