# Peer review of "Advanced Techniques for Pilotis RC Frames Seismic Retrofit: Performance Comparison for a Strategic Building Case Study"

_buildings, doi:10.3390/buildings10090149_

Round 1
Reviewer 1 Report
This manuscript investigates four different seismic retrofit schemes for Pilotis RC frames. The article’s objective is clear and of significance. However, there are some issues which need to be addressed before it’s accepted.
- The first two paragraphs could be simplified on the issue of the history of Piloits RC frames, which is not the main topic of this paper. On the contrary, the literatures about the comparison between the discussed four retrofit schemes should be expanded. At the same time, performance-based seismic literature reviews should be added.
- In section 2.1, please describe the origin of Figure 4’s backbone curve. Actually it is related to the rebars’configuration of the column and beam and the analytic model, please state the general consideration of the backbone curve.
- Depending on some circumstances, the damping mode of the nonlinear time-history analysis makes a substantial effect on the result especially for the energy dissipation device, which is not discussed in the paper. It is strongly suggested to be added in the content.
Author Response
The authors would like to thank You for your accurate work that helped improving the manuscript. In the attachment, the authors answer to the requests and show the main changes made to the manuscript.
Yours sincerely,
Eleonora Grossi

Reviewer 2 Report
This is a review of the manuscript “Advanced Techniques for Pilotis RC Frames Seismic Retrofit: Performance Comparison for a Strategic Building case study”. I think the article has carried out valuable work and is very useful for the research community in the related fields. However, there are some issues that must be clarified by the authors.
For research articles, abstracts should give a pertinent overview of the work. I suggest that the ABSTRACT provide the following elements: (1) Background: Place the question addressed in a broad context and highlight the purpose of the study; (2) Methods: Describe briefly the main methods or treatments applied; (3) Results: Summarize the article's main findings; and (4) Conclusions: Indicate the main conclusions or interpretations.
INTRODUCTION
Explain how this paper differs from the related ones published in the technical literature.
The aim should be clearly define in the introduction.
The authors can highlight the usefulness of the study in the practical applicability.
The authors should discuss the scalability of the proposed approach.
Highlight the novelty and originality of this study.
CASE STUDY
Please explain in more detail to the reader how the model was build taking into account all the parameters involved in this model. Could you clearly distinguish your contribution and the software tool use?
The analysis is shallow and there is a lack of critical discussion and conclusions. Please provide more information, details of the graphic representations, as well as provide a critical analysis of the results illustrated graphically in the figures.
Author Response

(The authors gave the same response as above.)
